# Emergency Digital Teaching during the COVID-19 Lockdown: Students' Perspectives

**Rolando Gonzalez \*, Hanne Sørum and Kjetil Raaen**

School of Economics, Innovation and Technology, Kristiania University College, 0153 Oslo, Norway;
hanne.sorum@kristiania.no (H.S.); kjetil.raaen@kristiania.no (K.R.)
**\*** Correspondence: rolando.gonzalez@kristiania.no

**Abstract:** This paper presents a qualitative study of the experience of students of the shift from face-to-face learning to online learning during the COVID-19 lockdown in Norway. Detailed inputs were collected from 200 university students enrolled in a bachelor's degree in information technology in Norway through an online survey. Their responses were categorized into three main themes: the teacher's role, the life of a student, and digital learning. We found that, surprisingly, the students felt that the shift to digital learning had positive effects on their lives, such as the availability of more time for study, study flexibility through recorded lectures which could be reviewed repeatedly and anytime, and more time to pose questions. However, some students also pointed out negative effects such as more distractions, lack of structure, and a perceived invasion of privacy when required to turn on their cameras. The students valued the use of high-quality technical equipment as well as student engagement during online lectures, but also freedom of choice to participate.

**Keywords:** digital learning; COVID-19; higher education; student perspective

## 1. Introduction

In March 2019, Norway, along with large parts of the world, was shut down due to the coronavirus disease 2019 (COVID-19). As a result of the pandemic and subsequent shutdowns, the landscape of higher education institutions underwent major changes [1]. In a matter of days or weeks, educational institutions had to transition to online teaching and choose which digital tools their lecturers would use to continue offering education to their students. This necessitated new types of technology infrastructure as well as support and guidance for educational staff who had neither used digital tools to deliver lectures nor taught online before. Indeed, this was a significant transition for teachers, who not only suddenly had to use digital tools but also to change their teaching plans. However, students also experienced significant changes. From sitting side by side with classmates and having physical interaction with the lecturer, student assistants, and peers, students have had to sit for long hours at home watching lectures on their screens and working in isolation.

The sudden transition from classroom education to digital education has been labelled emergency remote teaching (ERT) [2,3]. The term was coined to distinguish it from traditional online teaching, where the institution and the lecturers plan for online delivery ahead of time. Although many students have access to the internet at home through their mobile phones and other digital devices, there are other factors that make digital teaching and learning challenging and demanding. As mentioned, there is a marked difference between being in classrooms, auditoriums, libraries, the canteen, and other places in school with lecturers, student assistants, and peers, and studying alone at home in the living room or bedroom. While this is in many ways a challenge, however, this has also opened up some opportunities and positive experiences. People have learned to utilize digital tools to communicate in a professional context, such as to conduct meetings. Organizational meetings have become more efficient, as the participants do not have to travel for a long

time and incur travel and other expenses. Such transition has also shown us that we are more adaptable than we think.

Although physical human contact has been significantly reduced, we have found other ways and new ways to communicate and interact with each other. Much of this experience, we are likely to bring into the future and will likely affect how we will act in the coming years. For example, they will likely make communication between colleges, lecturers, and students more efficient and make each party more available. Human contact is important, but in exceptional situations, we must find solutions that work. Through the pandemic, we have gained a lot of useful experience in a short time.

While online courses and learning over the internet have been considerably studied over at least a decade, studies on them have significantly increased during the COVID-19 pandemic. This is partly because digital teaching during the pandemic differs from traditional digital teaching due to the limited time available for the preparation of both teachers and students [4]. What has piqued our curiosity is students' experience of being involuntarily online students in a time of much uncertainty. We had many questions regarding how the lockdown has affected students. Because we are working in higher education, that is where we focused our efforts. We summarize these questions in the following main research question:

How are higher education students experiencing digital teaching and learning during the COVID-19 pandemic?

To answer such a question, this paper draws on qualitative data collected through an online survey of bachelor's degree students in Norway. We had published the quantitative results of the survey [5] but not the qualitative results because of the space restrictions of the publication and because qualitative and quantitative results are very different in nature. This paper presents the qualitative results. We will later present a thematic analysis of the student responses to understand their experience of digital teaching and learning during the COVID-19 lockdown.

This paper is divided into six parts. In Section 2, we review the relevant literature. In Section 3, we describe the method used and the analysis performed. We present the findings in Section 4 and discuss them in Section 5. Section 6 concludes this paper and gives suggestions for future studies.

## 2. Background

The COVID-19 pandemic has significantly affected education worldwide. In a short time, the curriculum has been forced to be delivered in an online format. This has been a challenging process for the people involved [6], including teachers and students. Although digital teaching is not new, there has been a renewed focus on it with the onset of COVID-19. Students and staff who had originally signed up for on-site education were suddenly sent from the classroom to Zoom or other digital platforms. In contrast to the now traditional alternative known as online education, this sudden move to online learning is described as emergency remote teaching (ERT) [2,3]. Hodges et al. described the difference as follows:

> Typical planning, preparation, and development time for a fully online university course is six to nine months before the course is delivered. Faculty are usually more comfortable teaching online by the second or third iteration of their online courses. It will be impossible for every faculty member to suddenly become an expert in online teaching and learning in this current situation, in which lead times range from a single day to a few weeks. [3]

ERT has brought about many and varied experiences, some positive and others more challenging. Among them are the experiences of silence, loneliness, and not being able to meet those whom one wants to meet daily. A study [7] explored how the pandemic affected loneliness across population subgroups in Norway. Data were collected through an online questionnaire in June 2020. The general loneliness was stable or fell during the lockdown. However, some subgroups, individuals, and older women reported slightly increased

loneliness during the pandemic. The results of the study indicate that Norwegians seem to have managed the lockdown without an overwhelming increase in loneliness.

Moving on to the impact of COVID-19 on teaching students in higher education programs, Hjelsvold et al. [8] conducted a study in Norway on how teachers experienced the transition from location-based teaching (i.e., teaching face-to-face in physical environments) to online teaching (i.e., teaching through online platforms such as Zoom) during the lockdown. The findings showed that almost every teacher in the field of computer science experienced a positive change. However, the main challenge was related to pedagogical concerns. A study that continued the focus on the teacher perspective [9] yielded similar results. The teachers were mostly content with the tools and their handling of them; however, they saw challenges in inducing the active involvement of students and in conducting two-way communication with them. Various forms of stress were also mentioned. The findings from the previous studies [8,9] are interesting to consider from the perspective of Mittal et al. [10], who looked at performance expectancy (PE) and effort expectancy (EE) as factors that influence teachers' willingness to adopt a system. On the one hand, the teachers did not seem to have issues with the technology. The technology for delivering lectures is not complex and is necessary during the lockdown; that is, the PE should be high and the EE should be low. On the other hand, using technology to deliver lectures while maintaining pedagogical quality seems to be a challenge.

In 2020, Raaen et al. [11] conducted an online survey among students enrolled in a bachelor's degree in IT program capstone project. As a result of the pandemic, the students had to move their working space and collaboration into digital environments in a short time. That study showed that from a student perspective, this sudden change had a significant perceived negative effect on collaboration, communication, and results, an important reason being that had the lockdown not happened, they would have been working together in teams. However, the outcome measured with the grades given to the students implied that the students were unaffected by the situation. Consequently, the students felt affected by the lockdown, but in practice, they handled the stress well. Zawacki-Richter [12] conducted a study in Germany and looked at the effect of COVID-19 in light of ERT. It showed that while acceptance of e-learning tools had been slightly declining before the pandemic, the demand for digital innovations is expected to increase in the future. In other words, the pandemic will have a positive effect on digital innovations in university teaching in Germany. This may also be the case in other countries, such as Norway.

Klapproth et al. [13] performed a study in Germany after the switch to distance teaching due to COVID-19 that showed that teachers experienced medium to high levels of stress due to the situation. Most of the respondents experienced technical barriers, though most of them felt able to handle the stress. Furthermore, male teachers experienced significantly less stress than female teachers. In the context of digital teaching, Castelli and Sarvari [14] found that 90% of the students in their study did not turn on their cameras during synchronous lectures. The students ($n = 276$) were asked in a survey why they chose not to turn on their cameras. At the university where the data were collected, there was a policy that made it optional for students to turn on their cameras during online classes but encouraged students to do so. The students' reasons for not turning on their camera were, among others, concerns about their appearance and that the people in their household or physical location would be seen behind them; a weak internet connection; their belief that not turning on their camera was the norm; and their feeling that people were looking at them. Castelli and Sarvari state that one should not force the students to put on their cameras, as the student may have different living conditions which make it difficult. However, Castelli and Sarvari also propose to encourage it by explaining the benefits for both the students and the teacher, including the value of nonverbal cues in communication, building instructor-student and student-student relationships, and creating a warmer, closer, and more comfortable environment.

Gonzalez et al. [15] found that in an ERT situation, the digital learning environment must be scaffolded. Students need help in becoming independent and self-motivated; in developing a daily study routine; and in meeting and communicating with their peers. Their daily study routine is affected by the disappearance of the context switch that used to come from their going to school. In ERT, students' homes are their place of leisure, study, and—for students also working from home—work. Thus, student resistance to using video, sound, and chats is a challenge. Not using these means of communication can quickly become the norm, which will hinder students from communicating with their peers, teaching assistants, and teachers. Students are aware that communicating with others is beneficial. However, their resistance to exposure stops them from making use of the possibilities afforded by technology. Some students even resist communicating fully in smaller groups such as for project exams. Regarding daily study routines, Gonzalez et al. found that students saw live lectures as important because such lectures gave them events to organize their studies around, as they studied before and after lectures. As also mentioned by Zhou and Zhang [16], students miss being able to meet their teachers and peers in the online setting. Zhou and Zhang's student subjects further disclosed that the lack of live events is a major barrier to their online learning. They also found that the hybrid learning mode was optimal, as the students on campus reported better support for their studies.

Abou-Khalil et al. [17] identified engagement strategies that students enrolled in higher education programs but who had low resources found effective in the context of emergency online learning. They found that student-content engagement strategies such as screen sharing and class recording were perceived as most effective. Those were followed by student-teacher strategies, such as question-and-answer sessions and reminders. Student-student strategies such as group chat and collaborative work were considered the least effective.

Beyond the purely academic, life itself has been affected by the lockdown. Jun et al. [18] looked at first-year students in Korea. They found that new students felt profound disappointment after having looked forward to university for a long time.

Students also had difficulty adapting. For some, all this turned into depression. Despite this, students found the learning activities meaningful, and those who focused on such thinking handled the situation better.

Baloran et al. [19] conducted a study among students ($n$ = 529) in higher education programs in the Philippines to understand the effect of COVID-19 on students. The findings showed that satisfaction with online teaching was significantly correlated with the engagement among online students. The findings further showed that the students who participated in the survey had the same degree of satisfaction with online teaching but had various levels of online learning engagement based on their year level. In terms of student engagement, Farrell and Brunton [20] conducted a qualitative study in which they followed 24 online students in Ireland for over a year. The results showed that there were several psychosocial factors that influenced successful online student engagement, including an engaging teacher and confidence or self-efficacy among the students. The study also showed that the most challenging aspect of being an online student was balancing studies with other activities, such as work and staying connected with family and friends. This showed, among other things, that there is a smaller difference between schoolwork and other activities during the pandemic. Many students experience these activities as overlapping, without clear distinctions, unlike before.

Tando et al. [21], in a study on facilitators and inhibitors of the adoption of e-learning by undergraduate students, investigated several factors such as PE and hedonic motivation (HE). They found that the students preferred online learning if they perceived it as beneficial for themselves. Thus, it is important to help students develop a habit of using e-learning frequently, and it is important to encourage students to engage with their peers and teachers through interactive digital functions such as the chat functionality and other functionalities based on gamification.

Peimani and Kamalipour [22] conducted a qualitative analysis of the perceptions of student learning during the COVID-19 pandemic. The classes and materials were a mix of synchronous and asynchronous. The students had weekly online reading and discussion seminars using Zoom as the main platform. A high 82.1% of them were satisfied with the online delivery of lectures and reading seminars, and 88.9% were satisfied with the delivery of discussion sessions. The students preferred (82.2%) live lessons over prerecorded lessons because they found the former more helpful. Recording the live lessons facilitated asynchronous learning, enabling the students to review lectures at their own time and pace.

The students could communicate both orally and through text but were more comfortable communicating textually. They were satisfied (85.8%) with their communication with their tutors but were less satisfied with their interaction with their peers (28.6%). Interacting with their peers was a challenge for them as it became more of a monologue, and "many students (with cameras off) were sidelined in the online sessions due to non-participation" [22] (p. 9). Only 50% thought students should be expected to turn on their cameras during live online sessions, which is an interesting contrast to the 78.6% who thought it would be helpful for their learning experience to switch on their cameras specifically for the online discussion session.

To identify predictors of success in online learning, Kovačević et al. [23] identified and statistically verified four key factors: positive experience with the chosen learning platform, motivation to learn in the situation, the importance attributed to learning achievement, and the students' level of digital competency.

To bring this topic further forward, we need to dive deeper into the minds of individual students to mine their thoughts and impressions. Consequently, we see the need for deeper qualitative work exploring the hows and whys of digital learning.

## 3. Methods

This paper describes a qualitative study based on an online survey. Its purpose was to gather insights into students' experiences of digital learning during the pandemic, in their own words.

### 3.1. Survey Design

The questions were developed based on the authors' collective experience in teaching at the higher education level. We focused on topics such as participation, recording of lectures, and general experiences linked to digital teaching during COVID-19. The survey consisted of both quantitative questions and open-ended questions so that the respondents could offer qualitative comments and fruitful insights. We strove for a straightforward design, with precise and clear questions. A pilot test was conducted in advance to ensure that the questions were understandable to the target group. After the pilot test, a few adjustments were made.

In this article, we focus on the qualitative findings from the open-ended questions in the survey because as has been mentioned, the quantitative findings have been communicated in a previous paper. The questions are presented below, followed by the number of responses to each question.

1. What do you perceive works well in live lectures in Zoom, and what do you perceive does not work well? ($n = 130$)
2. Why do you prefer the recording or non-recording of lectures? ($n = 130$)
3. Why do you participate little or a lot in live lectures using chats, voiced questions, video, and other participation modalities? ($n = 128$)
4. What would it take to make you participate more actively in the lectures, using chats, voiced questions, video, and other participation modalities? ($n = 101$)
5. What advice do you want to give teachers to improve their digital lectures? ($n = 97$)

### 3.2. Data Collection

We conducted an online survey among bachelor's degree students in information technology (IT) on their first, second, or third years of study. To contact the students, we presented our study concept to them during a lecture and gave them a link to the online survey questionnaire, while assuring them of full anonymity. Thus, participation was voluntary, and we aimed to contact all, approximately 600 students, in the program. The survey was conducted from January to February 2021 using SurveyMonkey and closed with 200 respondents.

### 3.3. Respondents

The survey respondents were bachelor's degree in IT students. Thirty percent of them were women, 69% were men, and 1% did not want to state their gender. The age distribution is as follows: 48% were 18–24 years old, 46% were 25–34 years old, 5.5% were 35–44 years old, and 0.5% were 45 years old or older. The respondents' year level in university also varied: 59% were on their first year; 12.5%, second year; 28%, third year; and 0.5% answered "other". As part of the introductory questions, we also asked whether the students had paid work alongside their studies, and 35% answered no, 29% worked 1–10 h a week, 29% worked 11–20 h a week, and 8% worked more than 20 h a week (the percentage doesn't total 100% because decimals are rounded up). We were also interested in whether the students had a suitable place to sit when attending digital lectures. The results showed that 78% always had a suitable place to sit, 20% had it only sometimes, and about 3%, never (the percentage doesn't total 100% because decimals are rounded up).

### 3.4. Qualitative Analysis

The data analyzed in this article came from the answers of the respondents to our open-ended questions. Even though the literature we earlier reviewed had pointed out certain aspects of the digital learning environment during the pandemic, we did not find sufficient literature on how students are experiencing digital lectures in Zoom. Hence, our study was explorative in nature.

As we received many answers to our open-ended questions and we are a team of three researchers, we needed a clear process for analyzing the qualitative data. The answers ranged from descriptive to what the students felt about the situation. To have a more structured analytical process, we chose thematic analysis based on Braun and Clark's [24] six-phase process and Gibbs [25]. Thematic analysis is a tool for the researcher to go through qualitative data in a more predictable manner and to gradually discover overarching themes, that is, to discover patterns. Braun and Clark defined six steps in the process of thematic analysis:

1. Familiarization with the data;
2. Generating the initial codes;
3. Searching for the themes;
4. Reviewing the themes;
5. Defining and naming the themes; and
6. Producing the report.

Although we created questions that focused on specific issues, we wanted to let the data speak for themselves as much as possible, as described by Braun and Clark and Gibbs, instead of us coming in with preconceived assumptions. However, we also acknowledge that coming in blank without any thoughts, meanings, and expectations is not possible, as we, as teachers, are involved in the situation that we are studying. We describe our use of the thematic analysis process in the following paragraphs.

We began by downloading all the responses and entering them into a text document that ended up 80 pages long. In the first phase, our goal was simply to familiarize ourselves with the data. We read and reread the responses while taking notes. We also agreed among ourselves that we should not form conclusions too quickly, that is, that we should not

attempt to come up with codes or themes by ourselves but that we should meet to discuss our notes and thoughts.

The next step was to create codes. For this, we made a table where all of us could add codes and notes to the codes as we reread the responses. After two rounds of rereading and adding codes, we began categorizing and merging equal codes to make it easier for us to go into the theme identification phase.

We formulated initial themes and refined them in steps by finding the bigger stories and patterns until we saw that the themes were sufficiently clear and unique. Finally, we used the themes as departure points for both our literature review and our further analysis and discussion in this article.

## 4. Results

This section is divided into three sections, which included sub-sections. Each of the sections are illustrated in Figure 1, to give an overview of the structure of our analysis. In each of the sections, we provide our findings, highlighted by citations of the respondents related to each of the three topics that emerged from our analysis.

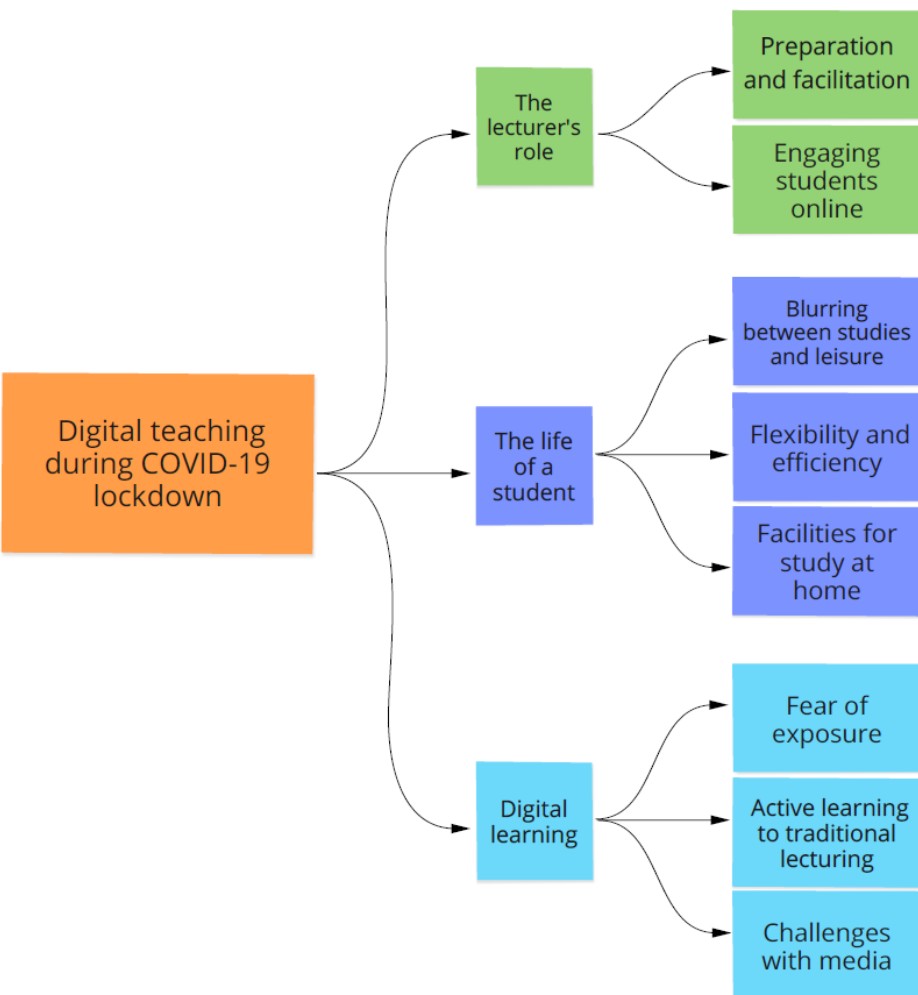

**Figure 1.** An overview of themes and codes identified in our analysis.

### 4.1. The Lecturers' Role

Digital teaching requires different and oftentimes greater preparation in advance of the actual teaching session compared to physical teaching. The increased preparation is partly due to technical facilitation through administration, technical competence, recording, use of tools, and other tasks. It is also important to engage the students by creating and

facilitating interactions and student activities. These require approaches different from those for physical teaching in a classroom setting.

### 4.1.1. Preparation and Facilitation

First, a digital teacher must have a different set of skills than a physical teacher. It is important to not only be a good teacher but also to facilitate a digital session that gives students a valuable experience and good learning. This is important both in terms of the content of the lecture and in relation to the actual implementation. Our findings showed that a range of factors play a role in this context. The teacher's way of using online tools and adapting to digital teaching influences the result and the students' experience.

The fundamental requirement for a teacher is to have access to equipment that works well, including computers, internet access, cameras, and microphones. Students should be able to focus on what is being said and not be disturbed by other elements such as poor sound quality and slow internet connection. Regarding this, one of the respondents wrote: "It's okay to follow, but the sound quality is rather poor for most people. An investment in good microphones from the school that the lecturers could make use of would have been better ". In line with this, another respondent said:

> The school should organize so that lecturers have a proper studio, at home or at school, where they can sit. Smaller groups ask everyone to have a camera and talk a little at the beginning of the lecture, before it becomes recorded. Get students involved and invested in the lecture—maybe some surveys along the way?

Moreover, if the teacher uses online tools such as Kahoot, they should also be well planned, and the teacher should be familiar with how such tools work. Furthermore, our findings showed that most of the students preferred that digital lectures be recorded and published afterwards. This requires advance planning by the teacher and the teacher's familiarity with publishing video recordings in the learning platform used. Many students favor recordings of lectures. One of them wrote: "[I] prefer to have recordings, very nice to be able to go through something difficult a few more times". Another student added: "It's worth gold. Lectures should be recorded regardless of whether it is home study or not. It's great to be able to review things several times or see later if things should come up that conflict with the lectures". The quality of the recordings should also be as good as possible.

Our respondents also mentioned that students prefer that the teacher answers their questions via the chat function during the session. For the teacher, however, this will be an "interruption" in the sense that the teacher will be "derailed" from the lecture. Regarding preparation, it is important that the lecturer has a plan for implementing this chat functionality effectively—whether he or she will answer questions continuously as they appear in the chat or collect questions after each lesson or between specific topics in the lecture. Related to this, a participant said: "It is important that breaks are taken so it is possible to have coffee. Do not go overtime unless it is said in advance". This emphasizes the importance of the lecturer planning the time well and sticking to the focus of the individual lecture. This must be done to respect not only the teacher's time but also the students' time.

### 4.1.2. Engaging Students Online

In many cases, it is easier for the teacher to engage students when they are physically in the same room. They see each other, and they can talk to each other, have a personal interaction with each other, and not least, observe the other's body language and how the other behaves. During digital sessions, such opportunities are often absent. Engaging students "through" the screen is harder, but our findings showed that the students have some preferences for engagement beyond the fact that the technical equipment must work optimally. First, it is important to have a good teaching plan that works and engages— among other things, through the use of tools and not just that the lecturer reads out the text on the PowerPoint slides presented. We also saw another key factor from our findings: that the teacher, during the lecture, encourages the students to be active. Active

participation may be asking the teacher questions and answering the teacher's questions, for example, using the chat functionality. However, this also requires the teacher to take the time to answer the questions that come in. If not, it will be perceived as meaningless for the students and can lower their motivation to actively participate in the lectures. Regarding engagement and participation during online lectures, one of the respondents wrote: "Things like, for example, Kahoot can make the lecture a little different and more captivating", but another respondent was somewhat more passive and wrote, "[I] have no advice. Understand that engagement is not easy to convey through a computer screen". Yet another respondent said: "Have assignments or exercises along the way that the student must do to help [keep up] his/her motivation. Live coding, where [students] can code together with the lecturer, is a great example".

### *4.2. The Life of a Student*

Due to COVID-19, the transition from physical to digital teaching came overnight. No one was prepared and many had to make changes, both in their private life and in their student activities. This created some opportunities that would otherwise not have been there, but also some challenges for many students.

### 4.2.1. Flexibility and Efficiency

Our findings showed that digital teaching provides greater flexibility and efficiency—flexibility because you can study "whenever you want", since lectures are prepared that are available 24/7. The students do not have to be present in a specific classroom at a given time to get the content of the individual lecture. This allows them to take more control of their daily lives, in terms of what to do at any given time. Some working students said this was good, among other things, as it helps them manage their work alongside their studies.

In terms of efficiency, the participants stated that digital teaching, as opposed to physical attendance in school, means less travel time to and from the campus. This is especially noticeable for those who have a long journey and spend a lot of time on trains, buses, and other public transportation. Note that we have no student accommodations on-campus and that housing in the immediate vicinity of our campus is expensive for most students. One participant said: "I think this [online teaching] generally works well; I am a big fan of this. Getting to the lecture is easy when it is live [online], [and] it is easier to combine work and studies".

### 4.2.2. Blurring between Studies and Leisure

One student put it bluntly: "Zoom works well, but everything being digital makes me lazy". Without fixed attendance times in school and, to a greater extent, with much of the learning left to the students, they experience less distinction between studies and leisure time than before COVID-19. This is not always positive and can lead to a less structured daily life for the students. Digital learning, as an alternative to studying in physical locations in school, provides reduced human contact and reduced communication, such as opportunities to contact a supervisor, teacher, and others. The everyday interaction with fellow human beings is considerably limited, and this entails, among other things, greater isolation and time alone for the individual. As a result, it is more important than ever for the student to plan his/her own time and when different activities are to be performed within a day or a week. A keyword is structure in everyday life. This is not as easy for all students to realize.

### 4.2.3. Facilities for Study at Home

Since the lockdown of society occurred over a very short period of time, there was little or no time to prepare for home study. Consequently, during the pandemic, some students have experienced challenges related to living conditions and varying degrees of access to suitable premises to follow teaching and studying. There are also marked differences in

living conditions among students. Some rent or own an apartment, whereas others live at home with their parents or in a shared accommodation. One of the respondents wrote:

> I think when it comes to online teaching, it's okay that we have a recording to watch in case you were not able to be 100% present at the lecture. If you are sitting in a room at school, at least there are [only a] few distractions. At home, there may be neighbors renovating, building right outside your window, etc. And it's generally harder to focus while at home.

Another respondent wrote:

> The disadvantage is that it is not always suitable to have a lecture on Zoom at home and is often more difficult to follow due to all the disturbances around. The sofa is not a place you want to sit and do schoolwork [in].

This testifies that students experience possible disturbances during digital teaching and that it is easier to focus and concentrate during face-to-face teaching.

*4.3. Digital Learning*

The digital environment in this context consists of several tools, such as Zoom, Discord, and Slack. The first is a video conferencing tool, and the last two are digital social platforms where students can get help from student assistants. Zoom offers digital meetings where all participants can participate via video, sound, and text chat. It is the main tool for delivering lectures to the participants in this study. Zoom also offers a function called breakout rooms, where the participants are split into groups of any size set by the teacher to enable discussions.

4.3.1. Fear of Exposure

Even though the digital tools make full participation possible, the students were reluctant to participate especially with video and voice during a lecture. The students would not turn on their cameras and would rarely, if ever, use voice. Text chatting during a lecture was more acceptable and was even seen as lowering the psychological threshold for asking a question for some of our students. However, several students thought it uncomfortable even to use the chat functionality out of fear of asking questions that may make them seem dumb. In digital social platforms, some students are reluctant to write posts that everyone can see, for example, to ask their peers and student assistants questions.

Breakout rooms, which could make discussions possible during a lecture, were disliked by the students. It was awkward for them to be with others in such a setting, especially if they did not know the others from before, something that would happen often since the teacher would assign random groups. Hence, although the tools made communication possible, this possibility was not used to its potential. A consequence of the use of digital tools was that some students become increasingly isolated as the semester progressed.

It is difficult to say how the students started being resistant to communicating digitally. One of the respondents said:

> The reason I wasn't so active was that most of the others [who] attended from the start [had] a passive mood. [I] felt a bit stupid [to be] the only one asking questions, and most seemed uncomfortable in breakout rooms.

Another respondent stated:

> It would have helped to know [with whom] I went to class, that there was room for asking, talking, discussing. Breakout rooms seem awkward, and many feel [they are] uncomfortable. [They work] against [their] purpose when many don't want to talk, turn on their camera, or participate.

An additional respondent pointed out: "Never make students discuss in breakout rooms. [It] will not happen. In 99% of the time, it will result in 5+ students sitting still, not saying a word, until the time is over". Technology is in place, but there are strong forces at play within the culture of the digital learning community that hold the students back.

### 4.3.2. From Active Learning to Traditional Lecturing

Some students perceived the teachers as reacting negatively to how they refused to turn on their cameras when encouraged to do so. Our respondents expressed their understanding of both the teacher's frustration and that it was unnecessary. According to them, the interactiveness declined strongly as time passed, with some teachers reverting to the traditional lecture style with monologues. This was understood to have been a consequence of the low activity level among the students in the digital lectures.

The respondents said they want more interactiveness, although few of them actively participated, and they noticed how the number of questions from the teacher has gone down and that dialogues between the teacher and the students no longer happened. At worst, the students perceived a live session as like watching a prerecorded video when the teacher did not include any form of interactiveness. A respondent spoke about the activity level in some of the lectures: "There's too little [interactiveness]. To just sit there and listen to somebody talk is not motivating". However, students also found the live sessions important because they give them the possibility of asking questions and the feeling of "being in school". Live lectures are preferred to the use of prerecorded lectures. As a respondent explained: "Sometimes, there are just prerecorded videos, and that is even less motivating, because then, I would rather find more engaging videos on the same topic on YouTube". Still, other students reflected on how digital lectures could not substitute for physical lectures and that they missed the feeling of truly being in school. A respondent who wished for increased interactiveness proposed to work during a lecture since the digital lecture is " . . . not very interactive and we learn much less by doing exercises on our own afterwards. We should have an arrangement where we also could participate, that we have assignments and tasks together". Breaking up lectures with small work sessions could have made for more interactive sessions.

### 4.3.3. Challenges with Media

Several technical and non-technical issues arose in the digital lectures. The most common issue was the quality of the sound in the lectures. A number of students found the sound quality problematic and referred to some teachers not having a good enough microphone. They added that background noises during a lecture could be disturbing, such as from children or animals. A respondent said: "Something that has been up and down is the sound quality. [In] 90% of the cases, it works fine . . . but at other times, there are birds making noise". Another student commented that " . . . the teacher's microphone is of too low quality" and that " . . . there are [still] some teachers [who] use the internal microphone on the laptop. The internal microphone hurts the ears of those listening, it records all the sounds in the room, and there's a lot of echo".

Bad habits of some lecturers, such as saying "uhm" or saying certain phrases or words repeatedly, became more pronounced. A student stated: "Some lecturers have bad habits [that] are magnified when they are the only person you see on screen. 'Umm', 'like', 'right', 'you know', etc. This can make it difficult to follow along when one notices this". Students also became very aware of how clear or unclear the teacher's pronunciation was, and how it would vary if the teacher spoke too slowly or too fast. Several of our respondents reported that they found it more difficult to focus on a digital lecture due to several factors such as distractions at home, thus returning to the issue of not having the context switch between leisure and study; and issues concerning sound would make this even worse. Also affecting focus was that some teachers forgot to give breaks, ending in too long and tiresome stretches of lectures. Our students are used to breaks approximately every 45 min, so requests for more frequent breaks refer to this baseline.

Another more pronounced issue for the students was their peers' use of the chat function during a lecture. On the one hand, they found the chat a good option for asking questions, but on the other hand, they found that many of their peers would spam the chat with unnecessary comments or questions they should have been able to find the answer to on their own. Moreover, the chat function in Zoom gives notifications and pop-ups

when somebody comments, and this was distracting for some of the students. The students also noted how even the teacher would get distracted by the chat and notifications and how this stopped the lecture because the teacher would have to read the messages, thus disturbing the flow of the lecture. However, in contrast to the negative comments on the use of the chat, there were several positive comments about the students being able to contact the teacher and answer peers' questions, and how this could have a positive effect on the interactiveness in class.

A smaller group of respondents said they chose not to participate with sound or text because they were worried about disturbing the teacher or their fellow students. A further reason given for not wanting to use voice was that there were other people talking or there were other noises where they were sitting. Some respondents mentioned lagging and quality of internet access as issues, but they seem to have been minor for most of the respondents.

## 5. Discussion

We see that the survey responses can be split into three broad categories that are discussed separately in this section before some broader conclusions are drawn.

### 5.1. The Teacher's Role

When teachers hold a digital teaching session, it is very different from teaching students in a classroom setting. This requires further preparation and facilitation, including engaging students, which is also a key factor for successful learning. Prior studies [8] have investigated how teachers experienced the switch from face-to-face teaching in physical environments to online teaching. The results showed that almost every computer science teacher experienced a positive change. This may be because, in such a field, one is used to handling technology and various tools in a teaching context, compared to other fields where technology is less important. In addition to facilitation of technology, the findings from our study also revealed the importance of recording lectures, as they provide students opportunities to watch the content afterwards and replay the recording as many times as they want especially if there is a subject matter that they find difficult and want to review.

Moreover, the quality of digital lectures should be as high as possible. Therefore, it is important that emphasis be placed on technical equipment. Suddenly conducting teaching in a different arena than what one is used to introduces pedagogical challenges. The teacher's way of using online tools in digital teaching influences the result and the students' experience. The teacher must have access to equipment that works optimally (light, microphone, camera, etc.) so that the students can focus on learning. The importance of technical facilitation is clearly evident in our findings.

The respondents further highlighted the need for active student engagement. Examples given were the teacher's use of digital tools (e.g., Kahoot) and encouragement of engagement among the students during lectures. Regarding this, previous research related to digital teaching has shown that there are challenges from a teacher's perspective, such as in relation to actively engaging the students and establishing two-way communication during online lectures [9]. In most cases, the teacher talks, and the students listen silently. Previous studies [19] have shown that engagement among online students was correlated with satisfaction. This shows that engagement is an important aspect of the experience associated with learning and satisfaction with the teaching.

While students want teachers to facilitate student engagement, prior research has shown that students do not turn on their cameras during online lectures [14] and therefore, in many cases, contribute to reduced engagement. In some contexts, it probably makes sense that students have not turned on the camera; but cases in which the teacher encourages it are different. From a student's point of view, it is sometimes easy to make demands about how a teacher should behave and at the same time, be passive and hide in the crowd with fellow students. From our findings, we also saw that some students want the teacher to

ask them for activity and commitment, while other students thrive best on being passive listeners and on not being forced to actively participate in online lectures.

In line with previous research [17,22], screen sharing and recording have been found to be effective in terms of learning among students, while question-and-answer sessions and reminders are also perceived as effective. Our survey respondents found recordings of lectures useful. The recordings mean that the students have access to lectures 24/7 and can use them, among other things, for exam preparations. It is therefore important that teachers record their lectures in subjects where recordings are appropriate to use.

### 5.2. Student Life during a Lockdown

We, as teachers, tend to view the learning experience based on what we are doing or telling our students to do. However, it may be argued that life itself and informal interactions between students are even more important for learning. Although we do our best to facilitate learning even during a lockdown, this informal part of studies is difficult for us to improve. From getting up in the morning, getting dressed, and commuting to campus, to going out drinking with fellow students, students have experienced profound changes to life itself during the pandemic that are important to how they handle the change.

Compared to the Korean students in a previous study [18], our students have somewhat lower expectations of life as a student. Many of them continue to live with their parents, and others move only short distances or go to college with old friends. These choices, combined with the lack of on-campus accommodations, also mean that university life is not as all-encompassing for these students as in the Korean case. Nevertheless, our students also miss the social aspects and the human interaction.

Furthermore, the blurring of lines between work and leisure demands a difficult balancing act. This is in line with previous studies [20], which found that the most challenging aspect of being an online student was related to balancing studies with other activities such as work and family life. Some students struggle to focus, while others feel more focused with fewer external distractions. The individual differences here are clearly important.

Interestingly, many students felt positive effects on their life of the pandemic changes. The reduction in commutes had saved them much time. The lack of social opportunities had increased the time available to them for studying. Combining this extra time with the flexibility of recorded lectures gives them great opportunities for focus and hard work. Even teachers who themselves blur the lines between work and leisure through heavy workloads contribute in many cases by being available for answering questions at any time. In contrast, some students feel that it is difficult to focus at home with all the distractions around them. The lack of a structure in such a flexible daily life is also difficult for many of the students to manage.

### 5.3. Digital Learning

Educational institutions strive to follow the tenets of active learning both online and in physical locations. Having active students participate in class, discussions, group work, and other forms of collaborative work make for better and deeper learning. Technology provides us several ways in which we can communicate and share information effectively, but we see in ERT that students are hesitant to engage fully, as would be most beneficial for them. For example, while communication through digital means would make it easier for students to communicate, communication seems to have been reduced dramatically overall in the digital learning space, judging by how students, especially in bigger classes, never turn on their cameras nor use voice, and some are even hesitant to write in the chat for all to see.

The reasons for students not participating fully in the digital learning environment in our study match those in literature [14,15]. The issue is exposure, which may be seen from two angles. In the first angle, students are wary about exposing themselves and how they look to others. Students are at home in their private quarters, such as in their bedrooms or living rooms, and may feel that it is unnatural to dress up for the occasion, as would be

normal if they were to travel to school to meet their peers. In the second angle, the issue of exposure may seem to some to be expressed as a fear of appearing dumb in front of others when asking questions and finding it awkward to speak in breakout rooms even in smaller groups, and even in important contexts such as in group exams. Our findings hint that an issue here is how well the students know each other. Some of our respondents wish they could know their peers before joining conversations with them through digital means. This leads us to a problem that is difficult to solve—for students to get to know each other, they must meet and talk with each other, but because they do not want to talk to strangers digitally, new relationships will not be initiated.

The best possible way that was seen to make students join discussions was breakout rooms, where students could meet in smaller groups of, for example, 4–6 students. However, our findings showed that in some cases, student groups ended up being silent for the entire allotted time, as was also reported by Gonzalez et al. [15] and Peimani and Kalamipour [22]. The students found this situation very awkward and uncomfortable. An important question is when and how this culture among the students started. Some answers indicate that they had been like that from the start. A few active students seem to have tried to start a new trend, but they quickly reverted as the group pressure to conform to the established culture of being invisible and silent became too strong. Some students further commented that they felt somewhat dumb for being the only one asking questions. In a physical classroom with many students, one may, of course, not have the most active students, but under the right conditions and as time passes, one may see an increase in participating students. In the digital learning environment in our study, we experienced that even the most active students do indeed fall back to inactivity. This fully reflects the phenomenon where the students express that they want more interactiveness but other students dislike it and do not partake in one of the most interactive forms possible. As found in [22], some students do think it would be a good learning experience for them to turn on their cameras. In addition to the issue of exposure, we see that the situation becomes so partly because of the students not knowing each other well enough or at all, as online environments offer fewer opportunities to engage with peers. To add other possible reasons, it may be asked if the teacher let the students prepare well enough to engage in a satisfactory manner, considering that in ERT, many activities could not be planned and adapted thoroughly to the new situation. These factors combined may cause students to feel less prepared and less confident to join group discussions.

The situation has not only affected the students but the teachers as well, according to the students' observations. The students noted how the teachers have made lectures less interactive, for example, lacking discussions between the teacher and the students or between peers in class. Some students noted how some teachers have reverted to monologues in class. Instead of engaging students in student-centered activities, some teachers have fallen back into exclusive instruction and transmission. We do not suppose it is their conscious decision to do so, but rather, a consequence of the situation.

The teacher may be hoping for dialogues, activities, and discussions in class, as we saw signs of in our findings. The students noted how teachers were trying to push for discussions in breakout rooms and asking the students questions during lectures in Zoom. It seems, however, that the teachers had given up after some unsuccessful attempts. After all, the teachers cannot force the students to turn on their cameras or use voice. Talking into a Zoom screen and watching black boxes with names instead of seeing students' faces is not the most motivating situation for the teacher. This entire situation is a prominent issue in the mentioned ERT situation. Students want interactiveness, but they may not be willing to fully engage. One may have the best of intentions to engage students, but the reality of the situation may not make it possible to achieve—at least not without knowing how the students may react and without planning how to prepare the students for such engagement.

On the bright side, even though the students are somewhat split, the chat functionality seems to have made possible some interaction between the students and teachers and among peers, as also found in previous studies [15,22]. On the one hand, the chat could

get spammed by irrelevant, distracting, and sometimes unnecessary questions, from the perspective of some of the students. On the other hand, this was the form of communication that was most used, as opposed to video and sound. A respondent in our study suggested having moderators in the chat, which may be a promising idea, especially in bigger classes. Moderators who could both moderate and answer questions to alleviate the work of the teacher in the chat could be beneficial. This may also help diminish the teachers' distraction due to too many messages in the chat, thus breaking the flow of the lecture. Students do not want to be distracted by the chat, but if they use it to ask questions, they expect, as also mentioned in [22], that the teacher is keeping track of their questions and answering them as they come.

Challenges with quality need to be addressed. Improving sound quality is relatively easy, and buying a good microphone for each lecturer should help. However, since the communication between teacher and student has been reduced, it could be that the extent of the issue did not reach the teachers or administrators as quickly as it should have. Sound and noise issues are important because we saw the respondents struggling with focusing at home, and this issue aggravated the situation. The teachers should also be aware of the quality of their articulation; their bad habits in speech such as pauses, use of superfluous expressions; and the speed at which they speak.

*5.4. Limitations*

The limitations of this study are typical of qualitative research. First, we asked the students how they have experienced digital learning during the COVID-19 lockdown, and how they answer may depend on what they emphasize, their subjective opinion, and what they best remember. In addition, the students in our sample belonged to a specific group—students enrolled in a bachelor's degree in an IT program in Oslo, Norway—and thus, they possibly have a different skill set and familiarity level with technology than other groups of students. Regarding differences between countries, it should also be noted that the students in this study do not live on-campus, as do students in some countries, but live at home, in their own apartment, or in student dormitories found in or around Oslo.

In addition, in the Norwegian context, Norway has had a relatively soft lockdown, in that the state and municipalities did not force their citizens to stay inside their homes, unless in specific cases of quarantine upon arriving in Norway from travel abroad. In general, the lockdown in Norway meant you could go outside as much as you liked. Visiting businesses and other homes was however severely limited at times.

Despite these local considerations, much of what we learned in this study should be internationally relevant. All activities were simultaneously moved from the campus to the home, which paralleled the experience across much of the world. While not living on campus, our students lived in shared apartments or dorm rooms provided by the student association. Some stayed with their parents during the pandemic. Thus, there is little reason to assume that our students had significantly different and better facilities for studying at home than do students from other places. Moreover, while cultures are different, life as a student is an important phase of people's lives across the world. Thus, we conclude that except for some details, our general results are relevant for most countries and cultures.

**6. Conclusions**

In conclusion, we saw that the students' experience in recording lectures was very useful in terms of flexibility and also so that the subject matter can be repeated and used in preparations for exams. Higher education students have many different requirements and needs. This requires that the lecturer understand the students well and facilitate interaction and interactiveness during digital lectures. Technical equipment must function optimally during digital lectures, and sound and video quality must not be distracting. Moreover, students prefer live lectures to prerecorded lectures because live lectures allow students to structure their day around such lectures. In addition, students find it difficult to turn on their camera during digital lectures; but when the camera and the sound are off, active

learning is reduced which leads to unfavorable student learning outcomes. We also found that frequent breaks are even more important online than during physical lectures.

### 6.1. Further Work

To fill a gap and increase understanding of digital teaching, as well as use findings from this study, there are several interesting studies that can add to the body of knowledge. The fear of exposing oneself with video and audio among students is something that recurs. It would be interesting to find out how to make the students more comfortable with the use of sound and image during lectures.

Our findings also show that many students are uncomfortable in group settings such as breakout rooms in Zoom. Discussions with fellow students often have a good learning effect and contribute to active learning. Therefore, one approach could be to investigate student involvement in online group discussions, which issues the students perceive as holding them back from communicating with their peers, and how to facilitate a comfortable setting from a student point of view.

### 6.2. Advise to Administrators

Many of the themes found in this work are complex and require dedicated work over time to improve. Fortunately, other findings are immediately fixable. We recommend the following:

- Consider recording and publishing online lectures.
- Provide each online lecturer with a professional-quality microphone and a quick course on how to use them.
- Make sure lecturers take frequent breaks after up to 45 min of sessions.
- Make sure to have a consistent and common set of tools and procedures for online lectures to reduce the workload of both the lecturers and the students.

Other issues, such as the passivity of the students and the lack of interaction, require more complex solutions that each institution and we, the research community, must continue working on together to deliver.

**Author Contributions:** Conceptualization, R.G., K.R. and H.S.; methodology, R.G., K.R. and H.S.; software, R.G., K.R. and H.S.; validation, R.G., K.R. and H.S.; formal analysis, R.G., K.R. and H.S.; investigation, R.G., K.R. and H.S.; data curation, R.G., K.R. and H.S.; writing—review and editing, R.G., K.R. and H.S.; visualization, R.G., K.R. and H.S. All authors have read and agreed to the published version of the manuscript.

**Funding:** This research received no external funding.

**Institutional Review Board Statement:** The study was conducted in accordance with the NSD cf. EUROPEAN PARLIAMENT AND COUNCIL REGULATION (EU) 2016/679 of 27 April 2016. We did not gather any personal identifiable information, and is therefore, according to NSD, not subject to ethical approval.

**Informed Consent Statement:** Informed consent was obtained from all the subjects of the study.

**Data Availability Statement:** This paper presents a qualitative analysis of free-text responses. These are personal in nature and include information on third parties. Thus, we cannot publish the data without violating privacy.

**Conflicts of Interest:** The authors declare no conflict of interest.

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
