# Peer review of "Emergency Digital Teaching during the COVID-19 Lockdown: Students’ Perspectives"

_education, doi:10.3390/educsci12030152_

Round 1
Reviewer 1 Report
The manuscript presents a qualitative study of students' experiences, which is interesting. The organisation is good and the results are presented clearly. However, in the introduction, I would expect to have more clearly stated problems and motivation, as well as clearly stated research questions. It would be nice to also state already in the introduction that the study is a part of another study, otherwise, it is a bit confusing when you first state it is a qualitative survey and then Likert scales are mentioned.
Please, check English language too, in some places it needs editing.
Author Response
We would like to thank you for having read through our article and given us good comments to improve it. We have created a table with your comments, and how we have met the suggestions for improvement.
All the best
Authors
Reviewer’s comments |
Changes made by us to meet the review |
However, in the introduction, I would expect to have more clearly stated problems and motivation, as well as clearly stated research questions. |
We clarify the purpose of the article in the following research question: How do students in higher education experience digital teaching and learning during the Covid-19 pandemic? |
It would be nice to also state already in the introduction that the study is a part of another study, otherwise, it is a bit confusing when you first state it is a qualitative survey and then Likert scales are mentioned. |
Done. References to quantitative data removed, because these are not used in the paper. Sorry for the confusion. |
Please, check English language too, in some places it needs editing. |
Edited for clarity and language. Please let us know if there is additional need for proofreading. |
|
|
Reviewer 2 Report
Dear Author,
Thank you for the opportunity to read your paper. I have the following comments which shall enhance the publication prospects of your paper.
- The paper, in its current form, lacks novelty. There already have been a number of papers published in online teaching / learning specifically in the context of COVID-19 pandemic. The novelty / need / significance of the paper need to come out strongly.
- Check this sentence in the abstract “From our study we see show some clear trends and patterns in students perception of digital teaching during lockdown”. Grammar needs to be corrected.
- Please check this: “In the search for an answer, this paper draws on qualitative data collected through an online survey among Bachelor students in Norway”. Do you mean a student who is a bachelor? Or students pursuing bachelor courses / programs? Rephrase accordingly.
- The literature review (LR) section is more like a collection of executive summaries / abstracts of previously published papers. This is not what literature review is. LR is to identify methodological shortcomings/ limitations in existing research which led to research gaps. It is these research gaps that would be the source of your research problem statement / aim of the study followed by research questions and research objectives.
- Some papers that may be added to strengthen the LR section:
- Mittal, A., Mantri, A., Tandon, U. and Dwivedi, Y.K. (2021), "A unified perspective on the adoption of online teaching in higher education during the COVID-19 pandemic", Information Discovery and Delivery, ahead-of-print No. ahead-of-print. https://doi.org/10.1108/IDD-09-2020-0114
- Tandon, U., Mittal, A., Bhandari, H. and Bansal, K. (2021), "E-learning adoption by undergraduate architecture students: facilitators and inhibitors", Engineering, Construction and Architectural Management, Vol. ahead-of-print No. ahead-of-print. https://doi.org/10.1108/ECAM-05-2021-0376
- In the methods section the survey form (instrument) development needs to be described in detail. The selection of the sample / sampling design also needs to be described in detail.
- You may consider shifting section 3.4 (limitations) towards the end of the paper.
- The section 5.4 ((final notes) is not needed. This makes the paper verbose and repetitive.
- Add 4-5 recent articles from this journal Education Sciences (ISSN 2227-7102) to enhance the positioning of your paper with the journal.
Best wishes.
Author Response
We would like to thank you for having read through our article and given us good comments to improve it. We have created a table with your comments, and how we have met the suggestions for improvement.
All the best
Authors
Reviewer’s comments |
Changes made by us to meet the review |
The paper, in its current form, lacks novelty. There already have been a number of papers published in online teaching / learning specifically in the context of COVID-19 pandemic. The novelty / need / significance of the paper need to come out strongly. |
We highlighted some findings that have not been clearly articulated yet. Maybe the most surprising among these are: -Technical quality of audio and video. -The need for breaks. |
Check this sentence in the abstract “From our study we see show some clear trends and patterns in students perception of digital teaching during lockdown”. Grammar needs to be corrected. |
Rewritten whole section. |
Please check this: “In the search for an answer, this paper draws on qualitative data collected through an online survey among Bachelor students in Norway”. Do you mean a student who is a bachelor? Or students pursuing bachelor courses / programs? Rephrase accordingly. |
Rewritten whole section.
|
The literature review (LR) section is more like a collection of executive summaries / abstracts of previously published papers. This is not what literature review is. LR is to identify methodological shortcomings/ limitations in existing research which led to research gaps. It is these research gaps that would be the source of your research problem statement / aim of the study followed by research questions and research objectives. |
Section now includes a clear statement of gaps in the literature. |
Some papers that may be added to strengthen the LR section: Mittal, A., Mantri, A., Tandon, U. and Dwivedi, Y.K. (2021), "A unified perspective on the adoption of online teaching in higher education during the COVID-19 pandemic", Information Discovery and Delivery, ahead-of-print No. ahead-of-print. https://doi.org/10.1108/IDD-09-2020-0114 Tandon, U., Mittal, A., Bhandari, H. and Bansal, K. (2021), "E-learning adoption by undergraduate architecture students: facilitators and inhibitors", Engineering, Construction and Architectural Management, Vol. ahead-of-print No. ahead-of-print. https://doi.org/10.1108/ECAM-05-2021-0376 |
Added Mittal et al. (2021) to related work, helping us to put other references on how educators make use of technology and what they experience are the challenges in a perspective of the theory in the article.
Added Tandon et al. (2021) to related work to add more findings on the driving factors to what helps students to choose to use e-learning. |
In the methods section the survey form (instrument) development needs to be described in detail. The selection of the sample / sampling design also needs to be described in detail. |
Method section rewritten to explain the process of creating questions and now includes a list of the questions used. Sample design clarified. |
You may consider shifting section 3.4 (limitations) towards the end of the paper. |
Limitations moved to end of discussion, as 5.4 |
The section 5.4 ((final notes) is not needed. This makes the paper verbose and repetitive. |
Section deleted. |
Add 4-5 recent articles from this journal Education Sciences (ISSN 2227-7102) to enhance the positioning of your paper with the journal. |
We have added: Peimani, N.; Kamalipour, H. Online Education in the Post COVID-19 Era: Students’ Perception and Learning Experience. Education Sciences 2021, 11. doi:10.3390/educsci11100633.
Kovačević, I.; Anđelković Labrović, J.; Petrović, N.; Kužet, I. Recognizing Predictors of Students’ Emergency Remote Online Learning Satisfaction during COVID-19. Educ. Sci. 2021, 11, 693. https://doi.org/10.3390/educsci11110693
Almazova, N., Krylova, E., Rubtsova, A. and Odinokaya, M. Challenges and Opportunities for Russian Higher Education amid COVID-19: Teachers’ Perspective. Educ. Sci. 2020, 10(12), 368; https://doi.org/10.3390/educsci10120368
Zawacki-Richter, O. The current state and impact of Covid-19 on digital highereducation in Germany. Hum Behav & Emerg Tech.2021;3:218–226.
|
|
|
Reviewer 3 Report
Introduction: It is necessary to explain the objective of the research and the gap that this study wants to cover.
Theorical Background: I recommend changing the title “Related Work” to “Theorical Background”. It is recommended to organize the theoretical part taking into account how the Covid affected and how they adapted since, for example, some references refer to the perceptions of the students and others to how the teaching methodologies were adapted. It is important to focus the references of the theoretical part on the objective of the research
Methololdology: It is necessary to include a table with the questions that were formulated and the theoretical foundation of the questions.
Could you explain the Thematic Analysis part more?
Limitations should be included at the end of the study and not in the methodology section.
The Results section is missing, although I understand that it is the Findings section. If a questionnaire with Likert scales was made, why are there no tables? If qualitative information has been analyzed, how has it been analyzed?
Implications: I cannot assess the implications because I do not understand how the analysis of the results has been carried out
Author Response
We would like to thank you for having read through our article and given us good comments to improve it. We have created a table with your comments, and how we have met the suggestions for improvement.
All the best
Authors
Reviewer’s comments |
Changes made by us to meet the review |
Introduction: It is necessary to explain the objective of the research and the gap that this study wants to cover. |
We clarify the purpose of the article in the following research question: How do students in higher education experience digital teaching and learning during the Covid-19 pandemic? |
Theorical Background: I recommend changing the title “Related Work” to “Theorical Background”. It is recommended to organize the theoretical part taking into account how the Covid affected and how they adapted since, for example, some references refer to the perceptions of the students and others to how the teaching methodologies were adapted. It is important to focus the references of the theoretical part on the objective of the research |
This is an exploratory study, and we have not focused on a specific theoretical foundation, thus we prefer “related work”. However, this section has been expanded and clarified to explain the gap we are trying to fill.
We have also modified the method section to clarify the exploratory nature of the study. |
Methodology: It is necessary to include a table with the questions that were formulated and the theoretical foundation of the questions. Could you explain the Thematic Analysis part more? If qualitative information has been analyzed, how has it been analyzed? |
Added the questions in the methods chapter. Expanded the explanation on what Thematic analysis and our process in doing thematic analysis, going from response data to codes and themes.
|
Limitations should be included at the end of the study and not in the methodology section. |
Limitations moved to end of discussion. |
The Results section is missing, although I understand that it is the Findings section. If a questionnaire with Likert scales was made, why are there no tables? |
Our article is based on the qualitative data from a survey consisting of both quantitative and qualitative data. The quantitative data was presented in its own separate paper, while our article here presents the qualitative findings. We have made this more explicit in this article. |
Implications: I cannot assess the implications because I do not understand how the analysis of the results has been carried out |
We hope the above changes clarified the implications. |
|
|
Round 2
Reviewer 2 Report
Dear Author(s),
The revised paper is a substantially improved version. However, I have the following comments:
(a) Please share the details of the Pilot Study (method, process, sampling, timeframe etc.)
(b) What is the justification for the sample size of 200? Is the pilot study sample included in this?
(c) Towards the end of the paper you need to highlight in 30-40 words how your findings may be generalized in other geographical contexts.
(d) An additional round of proofreading shall help.
Best wishes for your research.
Author Response
We would like to in this second round thank you again for having read our article and given us good comments to improve our article. We have created a table with your comments, and how we have met the suggestions for improvement.
All the best
Authors
Reviewer’s comments |
Changes made by us to meet the review |
Please share the details of the Pilot Study (method, process, sampling, timeframe etc.)
|
We have clarified this in the text. |
What is the justification for the sample size of 200? Is the pilot study sample included in this?
|
We have included more explanation around this. We made the survey available to around 600 students and 200 of these responded. |
Towards the end of the paper you need to highlight in 30-40 words how your findings may be generalized in other geographical contexts. |
We have done so. |
An additional round of proofreading shall help.
|
Article was sent to external proofreading. |
|
|
Reviewer 3 Report
Thank you very much for the clarifications and answers. The article has improved. However, I still do not agree with the title "Related Work", which I suggest modifying.
The conclusions could be developed a little more, relating the results with implications for management.
Author Response
We would like to in this second round thank you again for having read our article and given us good comments to improve our article. We have created a table with your comments, and how we have met the suggestions for improvement.
All the best
Authors
Thank you very much for the clarifications and answers. The article has improved. However, I still do not agree with the title "Related Work", which I suggest modifying.
|
Changed to Background |
The conclusions could be developed a little more, relating the results with implications for management.
|
We have added a subsection for implications for management. |
|
|